# Pharmacokinetics of metamizole (dipyrone) as an add-on in calves undergoing umbilical surgery

**Daniela Fux**[1], **Moritz Metzner**[2], **Johanna Brandl**[3], **Melanie Feist**[2], **Magdalena Behrendt-Wippermann**[2], **Anne von Thaden**[4], **Christine Baumgartner**[3]*

1 Institute of Pharmacology and Toxicology, Clinical Pharmacology, University of Veterinary Medicine, Vienna, Austria, 2 Clinic for Ruminants with Ambulatory and Herd Health Services, Center for Clinical Veterinary Medicine, Ludwig-Maximilians-University of Munich, Oberschleißheim, Bavaria, Germany, 3 Center of Preclinical Research, Klinikum rechts der Isar, Technical University of Munich, Bavaria, Germany, 4 German Center for Neurodegenerative Diseases (DZNE), Munich, Bavaria, Germany

* christine.baumgartner@tum.de

**Data Availability Statement:** All relevant data are within the paper.

**Funding:** Magdalena Behrendt-Wippermann got a grant from the H. Wilhelm Schaumann Stiftung

## Abstract

This preliminary clinical investigation of the pharmacokinetic behavior of the main metamizole (dipyrone) metabolites 4-methylaminoantipyrine (4-MAA) and 4-aminoantipyrine (4-AA) in calves undergoing umbilical surgery is part of an already published main study. A single intravenous dose of metamizole was added to ketamine/xylazine/isoflurane anesthesia. Eight Simmental calves weighing 90 ± 10.8 kg and aged 47.6 ± 10.4 days received 40 mg/kg metamizole intravenously 10 minutes prior to general anesthesia. Blood samples were collected over 24 hours and analyzed for 4-MAA and 4-AA. Meloxicam was additionally given twice: 2.5 hours pre- and 20.5 hours postsurgically. The pharmacokinetic profile of 4-MAA was best fitted to a two-compartment model and was characterized by a fast distribution half-life and slow elimination half-life ($t_{½alpha}$ = 5.29 minutes, $t_{½beta}$ = 9.49 hours). The maximum concentration ($C_{max}$ 101.63 µg/mL) was detected at the first measurement time point 15 minutes after administration. In contrast, 4-AA showed fast, high and biphasic plasma peak concentration behavior in five calves (2.54–2.66 µg/mL after 15–30 minutes, and 2.10–2.14 µg/mL after 2–3.5 hours) with a $t_{½beta}$ of 8.87 hours, indicating a rapid distribution and subsequent redistribution from well-perfused organs. Alternatively, three calves exhibited a slower and lower monophasic plasma peak concentration (1.66 µg/mL after 6.5 hours) with a $t_{½beta}$ of 6.23 hours, indicating slow accumulation in the intravascular compartment. The maximum concentration and area under the plasma concentration curve (AUC) of 4-AA were lower than those of 4-MAA. This metabolic behavior supports our already published data on clinical monitoring and plasma cortisol concentrations (PCCs). Compared to those of saline controls, lower PCCs correspond to the $t_{½alpha}$ of 4-MAA. Data on $T_{max}$ and $t_{½beta}$ also match these clinical observations. However, further studies are required to assess the exact analgesic mechanism and potency of the metamizole metabolites in calves.

Germany (https://www.schaumann-stiftung.de/de/forderung-1764.htm). The study was additionally supported by Richter Pharma AG, Austria (https://www.richter-pharma.at/). The funders had no role in study design, data collection and analysis, decision to publish, or preparation of the manuscript.

**Competing interests:** Magdalena Behrendt-Wippermann got a grant from the H. Wilhelm Schaumann Stiftung Germany (https://www.schaumannstiftung.de/de/forderung-1764.htm), and the study was additionally supported by Richter Pharma AG, Austria (https://www.richterpharma. at/). The funding sources had no involvement in the study design, collection, analysis and interpretation of data, the writing of the report and in the decision to submit the article for publication. None of the authors of this paper have any financial or personal relationships with other people or organisations that could inappropriately influence or bias the content of the paper. There were no conflicts of interest due to employment, consultancy, patents, products in development, marketed products etc., and there are no restrictions on sharing of data and/or materials. This does not alter our adherence to PLOS ONE policies on sharing data and materials.

## Introduction

Umbilical surgeries are very frequent in calves and are often performed under general anesthesia using ketamine/xylazine and isoflurane, accompanied by pre- and postsurgical nonsteroidal anti-inflammatory drug (NSAID) application [1, 2]. However, whether the drug protocol efficiently eliminates perioperative pain in calves undergoing umbilical surgery is unclear. The alpha$_2$ adrenoceptor-agonist xylazine is a potent sedative, but does not provide sufficient and long-lasting surgical analgesia [3]. The NMDA receptor antagonist ketamine is a short-acting analgesic but is less effective in the control of visceral pain [4, 5]. The analgesic effect of isoflurane at the clinically used dose is negligible. Moreover, a sufficient reduction of surgery-related inflammatory pain by NSAIDs is unclear [2]. Nevertheless, adequate control of surgical pain is of high importance in addition to ethical issues, because the effective reduction of perioperative pain significantly influences surgical outcome and recovery [6]. In the sense of multimodal balanced analgesia, different substance classes should be combined to produce the best possible analgesia with the fewest possible side effects. In cattle, the use of only approved substances for food-producing animals limits this choice.

Metamizole, is an analgesic and antipyretic pyrazolone derivative from a group of non-acidic, nonopioid analgesics. Other common names for this active ingredient are dipyrone and novaminsulfone. In human medicine this drug was launched in 1920 by the German Hoechst AG. However, the rare risk of reversible but potentially fatal agranulocytosis led to the introduction of compulsory prescriptions for metamizole in Germany in 1987 [7]. Today, risk assessment and evaluation vary considerably from country to country. In many parts of the world, including most countries in the European Union (EU), Latin America, Israel and Russia, metamizole is the most popular nonopioid first-line analgesic and is sometimes even available over the counter. On the other hand, in some countries, e.g., the United States, the United Kingdom, Canada, Australia, Japan, Sweden, Denmark, Finland, and India this drug is not even on the market for use in humans. In 1977, the U.S. Food and Drug Administration (FDA) removed approval of metamizole for human products and required that marketing of the drug for companion animals cease in 1995 to stop use in food animals. In 2019, the FDA approved Zimeta$^®$ (dipyrone injection) for the control of fever (pyrexia) in horses, but not for animals intended for human consumption or food-producing animals, including lactating dairy animals, due to safety concerns for humans [8]. In Canada, metamizole is registered for use only on small animals and horses, whereas in the EU metamizole preparations are approved for use in food-producing animals (horses, cattle, swine) against disease states in which a positive influence by the analgesic, antipyretic, and spasmolytic effect of metamizole is to be expected. According to Table 1 of Commission Regulation (EU) No 37/2010 [9], the current maximum residue limit (MRL) for metamizole in food producing cattle is 100 µg/kg each in fat, liver, muscle and kidneys and 50 µg/kg in milk. The withdrawal time in cattle before slaughtering is 12 days for consumable tissue and 4 days for the dispensing of milk. This is also based on the European Agency for the Evaluation of Medicinal Products (EMEA) summary that the overall risk of agranulocytosis to humans from the ingestion of residues from treated animals was negligible [10].

Metamizole is a prodrug that is rapidly transformed into the main metabolite 4-methylaminoantipyrine (4-MAA) after oral and intravenous administration. 4-MAA is subsequently metabolized to 4-formylaminoantipyrine (4-FAA) and 4-aminoantipyrine (4-AA), which are acetylated to 4-acetylaminoantipyrine (4-AAA) [11]. These compounds are eliminated primarily by the renal route, and the analgesic effect of metamizole correlates with the plasma concentrations of 4-MAA and 4-AA in humans [12]. In vitro, both metabolites inhibit the enzymes cyclooxygenase (COX) 1, 2 and 3 [13–15]. Moreover, 4-MAA and 4-AA modulate the activity

**Table 1. Main pharmacokinetic parameters of 4-methylaminoantipyrine (4-MAA) and 4-aminoantipyrine (4-AA) following single intravenous administration of metamizole (40 mg/kg) in calves.** Values are means.

| Parameter | 4-MAA (n = 8) | 4-AA (fast metabolizers, n = 5) | 4-AA (slow metabolizers, n = 3) |
|---|---|---|---|
| $R^2$ | 1.00 | | |
| Lambda z (1/hour) | | 0.08 | 0.11 |
| $t_{1/2alpha}$ (hours) | 0.088 (5.29 minutes) | | |
| $t_{1/2beta}$ (hours) | 9.49 | 8.87 | 6.23 |
| $T_{max}$ (hours) | 0.25 | 0.25 | 6.50 |
| $C_{max}$ (μg/mL) | 101.63 | 2.66 | 1.66 |
| K10 (1/hour) | 0.88 | | |
| K12 (1/hour) | 6.40 | | |
| K21 (1/hour) | 0.65 | | |
| C0 (μg/mL) | 500.64 | 2.79 | 0.10 |
| Vss (mg/(μg/mL)) | 17.76 | 249.19 | 283.36 |
| V1 (mg/(μg/mL)) | 1.65 | | |
| CL1 ((mg)/(μg/mL)/hour) | 1.45 | 19.40 | 28.18 |
| V2 (mg/(μg/mL)) | 16.11 | | |
| CL2 ((mg)/(μg/mL)/hour) | 10.55 | | |
| Vz ((mg)/(μg/mL)) | | 248.30 | 253.45 |
| $AUC_{0-t}$ (μg/mL*hour) | 482.06 | 35.87 | 26.97 |
| $AUC_{0-\infty}$ (μg/mL*hour) | 570.64 | 42.53 | 29.28 |
| $AUMC_{0-\infty}$ (μg/mL*hour$^2$) | 7010.28 | 546.25 | 294.38 |
| MRT (hours) | 12.28 | 12.84 | 10.06 |

$R^2$, correlation coefficient; Lambda z, terminal phase rate constant; $t_{1/2alpha}$, distribution half-life; $t_{1/2beta}$, terminal elimination half-life; $T_{max}$, time of peak; $C_{max}$, peak plasma concentration; K10, rate at which the drug leaves the system from the central compartment (elimination rate); K12, rate at which the drug passes from central to peripheral compartment; K21, rate at which the drug passes from peripheral to central compartment; C0, serum concentration at time 0; Vss, volume of distribution at steady-state; V1, volume of distribution in the central compartment; CL1, clearance of the central compartment; V2, volume of distribution in the peripheral compartment; CL2, clearance of the peripheral compartment; Vz, volume of distribution based on the terminal phase; $AUC_{0-t}$, area under the plasma concentration–time curve; $AUC_{0-\infty}$, area under the plasma concentration–time curve extrapolated to infinity; $AUMC_{0-\infty}$, area under the first moment curve from zero to infinity; MRT, mean residence time.

of cannabinoid CB1 receptors and the transient receptor potential cation channel TRPV in rodents, which suggests analgesic functions of the metabolites by exploiting the endocannabinoid/endovanilloid system [16, 17]. A recent study in rats revealed indirect activation of the kappa-opioid receptor with metamizole administration, which implied the participation of the opioid system in metamizole-mediated analgesia [18]. However, the exact mechanism of the analgesic action of metamizole is not yet fully understood.

Drug metabolism and therewith the analgesic efficiency of metamizole might be affected by immature hepatic and renal functions in young animals [19]. Moreover, pharmacodynamic and -kinetic interactions with coapplied drugs might occur in principle in anesthetized animals. It was therefore interesting to verify whether conversion of metamizole to its active metabolites in calves occurs efficiently at all when metamizole is used under surgical conditions.

The manufacturer-recommended metamizole dose for cattle is 20–40 mg/kg every 8 hours, applied slowly and intravenously (IV). Dose and application recommendations are derived from pharmacokinetic parameters determined after administration of a single dose of metamizole IV and/or intramuscularly (IM) in several species, such as sheep [20], donkeys [21], pigs [22], horses [23], cats [24], dogs [25] or goats [26], but to our knowledge kinetic data in cattle can only be obtained from dairy cows after IV application of a single dose of a combination

product containing butylscopolamine and metamizole [10]. As these data cannot be applied to anesthetized calves treated with different narcotics, we preliminarily determined the metabolic behavior of 4-MAA and 4-AA after a single intravenous dose of metamizole in few calves within the scope of a more comprehensive study design. The effects of metamizole on intraoperative and immediate postoperative nociception in anesthetized calves have been published elsewhere [27] and were investigated under the same conditions and using the same drug doses.

## Materials and methods

This study was performed in compliance with the EU Directive 2010/63/EU for animal experiments and the German Animal Welfare Act (2018). All procedures were approved by the Ethical Committee for Animal Experiments of the Government of Upper Bavaria, Munich, Germany (Reference Number 55.2-1-54-2532-12-13).

The clinical, veterinary research study was conducted between August 2013 and July 2014. When the outside temperature dropped below freezing (end of October 2013 to mid-March 2014), the performance of the experiments was interrupted because of the increased risk of perioperative adverse events (mainly regulation of body temperature).

### Animals

Eight Simmental calves (one female, seven males) were purchased by the clinic from a local livestock market (Zuchtverband Miesbach, Germany) for this study. On the day of purchase, a short clinical examination by a veterinarian showed no health disorders apart from uncomplicated umbilical hernia. The average age of the animals was 47.6 ± 10.4 days, and the average body mass was 90 ± 10.8 kg. In the Clinic for Ruminants of the Ludwig-Maximilians University of Munich (Germany), the animals were housed in individual pens (igloos) with hearing, olfactory and visual contact with conspecifics at all times. The experimental animals were strictly isolated from regular clinical animals by spatial and hygienic measures.

The calves were fed whole milk three times a day and had free access to water, hay, a total mixed ration (50% corn silage, 25% calf grain and 25% calf muesli), and mineral licks. The calves stayed at the clinic at least 6 days before surgery. The inclusion criteria were based on clinically unremarkable findings by a veterinarian and an undisturbed general condition of the animal, including feed intake and blood analysis (complete blood count (CBC) without a differential) one day before surgery. The exclusion criteria were any signs of disease other than an uncomplicated hernia. CBCs without a differential were only examined when calves entered the clinic to ensure their suitability for the study. However, afterward, CBCs were not routinely performed if there was no concrete clinical suspicion. No complications (fever, peritonitis, surgical incisional infection) occurred during the study, and the animals were sold afterward to a local cattle dealer. This was possible because only substances, that are listed as allowed substance for food-producing animals in the European Commission Regulation (EU) No 37/2010 were applied to the calves. Therefore, slaughtering was still possible after awaiting the minimum mandatory withdrawal time for consumable tissue, which is listed in the European Commission Regulation for each allowed substance.

### Anesthesia and metamizole application

The implemented procedure is described in detail by Metzner et al. [27]. Briefly, one day before surgery (d-1), a catheter (14 gauge, 40-mm catheter; VasoVet; B. Braun Melsungen, Germany) was inserted into a jugular vein after sedation with 0.2 mg/kg xylazine hydrochloride IM (Xylazin; Serumwerk Bernburg, Germany). The catheter was used for blood sampling and drug injection. Before and after each blood sampling or drug administration through the

jugular venous catheter, it was flushed again with saline, its correct position was checked, and the neck region was checked for non-irritation.

On the day of surgery (d0), calves were given 0.5 mg/kg meloxicam IV (Metacam; Boehringer Ingelheim Vetmedica GmbH, Germany) 2.5 hours preemptively, which was repeated 20.5 hours postsurgically. Metamizole was applied IV one hour preemptively at the recommended dose of 40 mg/kg (Metamizol WDT, WDT, Garbsen, Germany). Ten minutes after metamizole application, the calves were sedated with 0.2 mg/kg xylazine IM, and anesthesia was induced with 2 mg/kg ketamine IV (Ursotamin; Serumwerk Bernburg AG, Germany) 20 minutes later. Calves were endotracheally intubated and maintained with isoflurane (Isofluran Baxter Vet, WDT, Garbsen, Germany) in 100% oxygen via a circular system (Sulla 808 with Ventilog 2; Dräger, Germany). End-tidal carbon dioxide ($PE´CO_2$) and end-tidal isoflurane (FE´Iso) were recorded by a gas monitor (PM 8050; Dräger, Germany) every 6 minutes starting 3 minutes after coupling the system to the endotracheal tube until the end of anesthesia, 60 minutes after skin incision. After a stabilization period, FE´Iso was maintained at 1.2–1.6% and oxygen flow at 10 ml/kg/minute. Animals were mechanically ventilated by intermittent positive-pressure ventilation (IPPV) using a tidal volume of 10 ml/kg, a positive end-expiratory pressure (PEEP) of 0.5 kPa (5.1 cm $H_20$) and an inspiration/expiration ratio of 1:2. The respiratory rate was adjusted to maintain a $PE´CO_2$ between 4.7 and 6.0 kPa (35 and 45 mmHg). Umbilical surgery (extirpation of an uncomplicated hernia) was performed in dorsal recumbence by two experienced surgeons. The surgery time (incision to end of suturing of the skin) was 52 minutes, and the anesthesia time (induction with ketamine to disconnection of the endotracheal tube from the anesthetic machine) was 90 minutes.

## Determination of 4-MAA and 4-AA

Blood was obtained immediately before (time 0) and 0.25, 0.5, 1, 1.5, 2, 3.5, 6.5, 9.5, and 24 hours after metamizole administration. Before each blood sampling, the intravenous catheter was flushed with a 0.9% sodium chloride solution, and 2 mL of blood was withdrawn and discarded. Subsequently, 2 mL of EDTA blood was taken, and plasma was isolated by centrifugation of the blood samples (10 minutes, 3363 × g) and stored at -70 ˚C until further analysis. Plasma was deproteinized by the addition of acetonitrile/methanol and subjected to liquid chromatography-mass spectrometry (LC-MS/MS) via isocratic reverse-phase high-performance liquid chromatography (HPLC) and subsequent electrospray ionization on a triple-quadrupole mass spectrometer (API4000, Sciex). Analyses were performed in the Medical Laboratory Bremen (MLHB), Germany, and validated by a spike-and-recovery study using calf plasma.

Method validation details:

| | |
|---|---|
| Internal standard | Clonidin |
| Intra-assay coefficient of variation, CV% (N = 9) | 4-AA: 2.5%<br>4-MAA: 2.0% |
| LOQ (limit of quantification; S/N = 10) | 4-AA: 0.1 µg/mL<br>4-MAA: 0.1 µg/mL |
| LOD (limit of detection; S/N = 3) | 4-AA: 0.02 µg/mL<br>4-MAA: 0.04 µg/mL |
| Linear ranges | 4-AA: 0.1–20 µg/mL<br>4-MAA: 0.1–50 µg/mL |
| Method | LC-MS/MS |
| Material | Serum or plasma |
| Stability | 3 weeks at 2–8˚C |

The inter-assay variance is mapped by the running controls of the MLHB. These are typically in the following range:

Interassay variation in coefficient of variation in %

| | Target low control μg/mL | Average found μg/mL N = 16 | CV % | Target high control μg/mL | Average found μg/mL N = 16 | CV % |
|---|---|---|---|---|---|---|
| 4-AA | 1 | 1.05 | 6.8 | 5 | 5.08 | 6.7 |
| 4-MAA | 1 | 1.02 | 7.1 | 5 | 4.8 | 7.4 |

**Pharmacokinetic analysis.** Based on the measured plasma concentrations of 4-MAA and 4-AA pharmacokinetic parameters were calculated by using *PKSolver2.0* software, an add-in program for pharmacokinetic data analysis in Microsoft Excel [28]. Different models were fitted to the data, and a two-compartment model (CA, IV bolus, 2 compartment) best described the 4-MAA kinetics ($R^2$ = 0.9975). Noncompartmental analysis (IV bolus) using the linear trapezoidal method was used to calculate the 4-AA kinetics parameter.

## Results

The main pharmacokinetic results are listed in Table 1. A median $C_{max}$ for 4-MAA of 101.63 μg/mL (range: 52.00–229.2 μg/mL) (Fig 1) was detected in the first blood sample 15 minutes after metamizole application (time point 0.25 hours). Subsequently, 4-MAA disappeared from the plasma in accordance with a two-compartment model, with a distribution half-life ($t_{½alpha}$) of 5.29 minutes and an elimination half-life ($t_{½beta}$) of 9.49 hours. Twenty-four hours after metamizole application, the plasma 4-MAA concentration was 3.28 ± 2.49 μg/mL. The

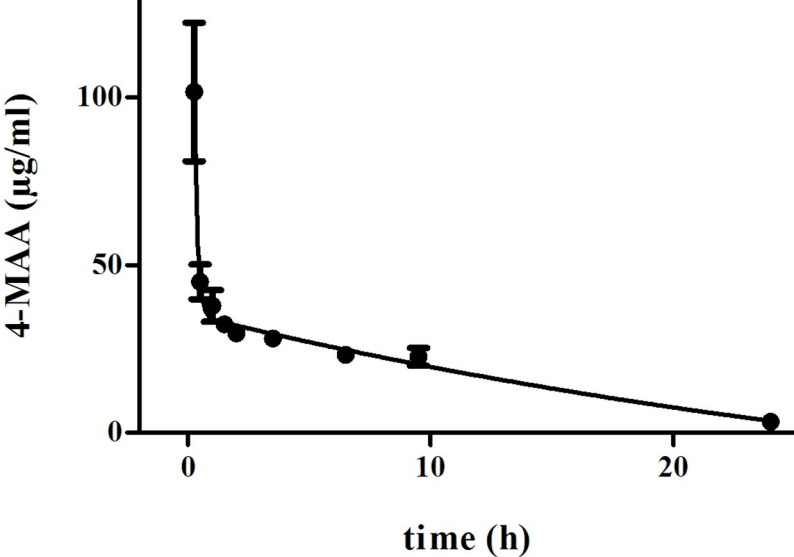

**Fig 1. Plasma concentration over time of 4-MAA after intravenous administration of 40 mg/kg metamizole in calves subjected to ketamine/xylazine/isoflurane anesthesia.** Values are the mean ± S.D. from n = 8 calves.

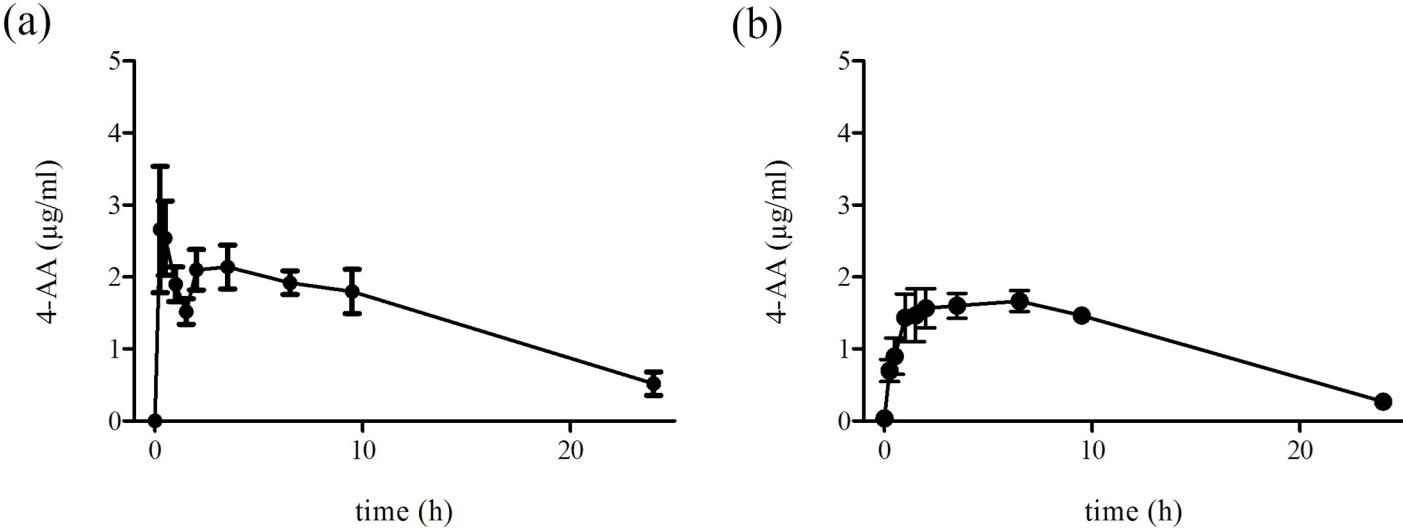

**Fig 2.** a. Plasma concentration over time of 4-AA after intravenous administration of 40 mg/kg metamizole in "fast metabolizing" calves subjected to ketamine/xylazine/isoflurane anesthesia. Values are the mean ± S.D. from n = 5 calves. b. Plasma concentration over time of 4-AA after intravenous administration of 40 mg/kg metamizole in "slow metabolizing" calves subjected to ketamine/xylazine/isoflurane anesthesia. Values are the mean ± S.D. from n = 3 calves.

individual $AUC_{0-t}$ values ranged from 407.45 to 800.15 µg/mL*hour. The calculated mean $AUC_{0-t}$ of 4-MAA was 482.06 µg/mL*hour.

4-AA was detected in all animals at all time points (0.25–24 hours). In five calves, the first plasma peak concentration was observed 15–30 minutes after metamizole administration (2.66 ± 1.96 µg/mL after 15 minutes and 2.54 ± 1.15 µg/mL after 30 minutes). In these calves, the plasma 4-AA concentration declined within 1.5 hours to approximately 43% but increased to a second peak 2–3.5 hours after metamizole administration (2.10 ± 0.63 µg/mL after 2 hours and 2.14 ± 0.67 µg/mL after 3.5 hours) (Fig 2a). The elimination half-life of 4-AA was calculated as 8.87 hours, and the calculated mean $AUC_{0-t}$ of 4-AA was 35.87 ± 6.8 µg/mL*hour. In contrast, three calves showed a monophasic increase in plasma 4-AA, which reached a maximum ($C_{max}$ 1.66 µg/mL) 6.5 hours after metamizole injection (Fig 2b). The calculated elimination half-life was 6.23 hours, and the mean $AUC_{0-t}$ was 26.97 ± 4.1 µg/mL*hour.

## Discussion

Although the pharmacokinetics of 4-MAA and 4-AA have been recently studied in various species after a single administration of metamizole IV and/or IM, e.g., in sheep [20], donkeys [21], pigs [22], horses [23], cats [24], dogs [25] or goats [26], this study preliminarily assessed 4-MAA and 4-AA kinetic behavior in eight calves under realistic perioperative conditions and simultaneously treated them with other drugs. Combinations of drugs, however, are commonly administered in the clinic, as many medical indications require pharmacological modulation of multiple targets. For example, adequate pain management after umbilical surgery in calves involves the coapplication of sedatives, narcotics and analgesics [1]. These eight calves were part of a more comprehensive study design including 26 animals in total. Data on clinical monitoring and plasma cortisol concentration (PCCs) from all 26 animals, including the saline control group without metamizole, have already been published [27]. Additional investigation of 4-MAA and 4-AA levels should clarify whether these putative main analgesic metabolites of metamizole are formed at all in anesthetized calves because of not yet fully developed

metabolic pathways and/or possible interactions with coapplied drugs. However, the metabolic behavior of metamizole in a realistic clinical setting is a limitation of our study because of the lack of a control group without presurgically administered meloxicam. Therefore, our kinetic data in calves should be classified as preliminary without comparison, based on which further research could be planned.

In the original standard anesthetic protocol of the Clinic for Ruminants of the Ludwig-Maximilians University of Munich (Germany) ketamine and xylazine constitute the intraoperative portion of analgesia and meloxicam mainly constitutes the postoperative portion, although it was already administered preemptively. Metamizole was tested as an add-on to improve intraoperative analgesia.

While metamizole is not uniformly available worldwide for food-producing animals due to the known human agranulocytosis risk, a recent survey from Germany including neutropenia showed a rare risk rate of only 1:1602 [29]. It was suggested that metamizole-induced agranulocytosis in humans is an immunologically mediated response and that stimulation of lymphocytes is a prerequisite. The Committee for Veterinary Medical Products concluded that the overall risk of agranulocytosis to humans from the ingestion of residues from treated animals was negligible [10]. Furthermore, data generated to support the approval of Zimeta® by the U.S. FDA [8] did not indicate that agranulocytosis occurred in horses treated with the drug. Therefore, we considered metamizole as a very suitable add-on drug for our calves.

In the overarching study design in which our calves were integrated, first the clinical monitoring parameters (especially heart rate (HR) and mean arterial blood pressure (MAP)) were investigated intraoperatively and compared in groups with and without metamizole as an add-on. To better verify the unconscious nociceptive reaction to painful stimuli during surgery, the PCC as an indicator of nociception was additionally measured. Briefly, the already published [27] PCCs showed no significant difference in baseline values before surgery (p = 0.84). During surgery, general cortisol release could be observed in all animals, but compared to that in saline controls (CGs), the increase in PCC was delayed and consistently lower in metamizole-treated calves (MGs). Afterward, the PCC remained elevated in the CGs, whereas at 150 minutes after skin incision a significant decrease was observed in the MGs (MG: 11.6 (8.4–16.5) versus CG: 39.1 (27.3–81.4) nmol/L; p = 0.0026). Overall, the mean PCC in the MGs was 10.9 nmol/L lower than in the CGs (p = 0.01). After 8.5 hours, the PCCs were equal again in both groups (CG: 19.71 ± 16.51 nmol/L; MG: 15.98 ± 7.88 nmol/L). These results fit with the observed preliminary metabolic data (elimination half-life ($t_{1/2beta}$) and $C_{max}$ of 4-MAA and 4-AA) of metamizole.

Although the first blood sampling in sheep, goats, horses and donkeys [20, 21, 23, 26] was performed after 15 minutes, the maximum 4-MAA concentration was calculated a few minutes after intravenous metamizole application with a $T_{max}$ of 0.08 hours (= 4.8 minutes). Assuming a similarly rapid onset of 4-MAA in calves, a higher plasma concentration and thus a $C_{max}$ at a relatively early time point is conceivable. Since this scenario cannot be excluded, the $C_{max}$ value determined in the present study after 15 minutes should be considered the "first point $C_{max}$". In comparison with other studies in which metamizole was injected IV, our chosen blood sampling time points were comparable. Only in studies in which metamizole was injected IM was a longer metabolization expected, and therefore sampling times up to 72 hours were added [20, 22].

Blood pH underlies age-dependent variations; specifically, starting with mild acidosis after birth, the pH value of blood increases in calves during the first month of life and reaches an adult level by the age of 24 weeks [30]. Calves in the present study were aged between 5.3 and 8.3 weeks (47.6 ± 10.4 days) and their measured arterial blood pH value was slightly acidotic (7.35–7.36) [27]. It is thus likely that age-dependent mild acidosis leads to fast metamizole

hydrolysis (esterase-dependent conversion) in calves, as described in humans [31]. However, the influence of oxamic acid metabolites that are formed by meloxicam [32], on the pH value of the calves is negligible, as clinically relevant metabolic acidosis by NSAIDs is only observed at toxic doses [33].

However, the relevance of differences in plasma peak concentration is limited without additional evidence of clinical efficacy because plasma concentrations need not automatically be proportional to the clinical efficacy. Furthermore, $C_{max}$ and $T_{max}$ are not yet evidence of an inhibitory clinical effect above an analgesic threshold (minimum amount required to achieve pain relief) or therapeutic endpoint. Our $T_{max}$ and $t_{1/2beta}$ data seem to match the clinical observations, as the behavior of 4-MAA fit best to a two-compartment model, which means that 4-MAA in calves is rapidly distributed into extravascular compartments, primarily highly perfused organs such as the brain and liver. Fast 4-MAA distribution into the brain may account for a rapid onset of analgesia triggered by the release of ß-endorphin and synthesis of endocannabinoid [11]; distribution into the liver indicates metabolization by hepatic enzymes. Whereas 4-MAA accumulation in the brain is difficult to prove, an increase in 4-AA plasma concentration indicates 4-MAA demethylation by enzymes of the hepatic cytochrome P450 (CYP) system. Interestingly, the generation of 4-AA occurred in five calves faster than in three other tested calves suggesting individual differences in metabolic liver activity. Indeed, 4-MAA demethylases belong to the family of CYP2B and CYP3A enzymes, which are known for inter-individual variabilities in expression and activity in cattle [34, 35]. Compared to that in adult cattle, however, 4-AA synthesis was slower overall in calves, which may be caused by immature activity of the two enzymes of the CYP monooxygenase superfamily CYP2B and CYP3A [36] or substrate competition, as ketamine and xylazine are also CYP3A substrates [37]. Human data show that meloxicam, or rather its 5-hydroxylation metabolite, is predominantly catalyzed by CYP2C9, with only a minor contribution of CYP3A4 [38]. Attributable activity and concentration differences in these biotransformation enzymes in the liver, specifically CYP2C9, have been identified in sheep, cattle, and goats [39]. Generally, CYP enzymes are species-specific. In cattle, tissue-specific mRNA expression of different CYP isoforms could be proven. The absolute quantification of liver mRNAs showed that CYP3A38 was the most abundantly expressed CYP3A isoform in bovine liver, followed by CYP3A48. Conversely, CYP3A28 (corresponding to abundant human CYP3A4) was expressed at levels <1% in different cattle breeds. Similar to humans, physiological factors such as age, sex and breed have been shown to affect bCYP3A expression and/or activity [40]. Although we have not investigated the influence of age, sex and breed in our study, it is known from the literature that breed causes more differences in drug metabolizing enzymes (muscle:body ratio) in calves than sex [41, 42].

The manufacturer-recommended metamizole dose for cattle is 20–40 mg/kg every 8 hours, applied slowly and IV. The $t_{1/2beta}$ of 4-MAA in anesthetized calves of 9.49 hours verified the supposed duration of analgesic action of at least 8 hours [15] after a single dose of metamizole. Compared with that in our anesthetized calves, the $t_{1/2beta}$ in humans (1.6–3.6 hours), sheep (1.45–3 hours), goats (0.72 hours), horses (3.34 hours) and donkeys (1.81 hours) is much shorter [20, 21, 23, 26]. However, it is important to only compare study results using the same administration route due to prolonged gradual drug release from the injection site to the vascular system after IM administration [20, 22, 23]. Additionally, it is important to only compare results based on pure metamizole formulations, as combination products (e.g., Buscopan compositum®) can vary due to pharmacokinetic interactions between the two active compounds that can affect metamizole metabolism or 4-MAA kinetics [21, 23].

As the disposition of 4-MAA includes both metabolism and renal elimination, reduced liver and kidney function may be associated with delayed 4-MAA elimination in anesthetized

calves. Whether the difference is species-specific or rather results from reduced hepatic and/or renal blood flow and/or function during anesthesia remains to be investigated. Indeed, the COX-2 inhibitor meloxicam is known to reduce renal blood flow, and slight renal hypotonia can be caused by isoflurane. As our calves were aged between 5.3 and 8.3 weeks, we can still assume a slight immaturity in their liver and kidney function. Calves undergo metabolic and digestive tract physical changes during the weaning process (in the dairy industry, these changes occur between 6 and 9 weeks of age). When they start to consume solid feeds (such as concentrate feeds) and the rumen starts to develop, a shift in hepatic function alters metabolite and enzyme levels in blood. Renal and hepatic function-indicating enzymes develop adult levels within 24 hours and 7 weeks of age [43]. Therefore, the results of calves can differ from those of adult dairy cows.

The PCC data of our five biphasic 4-AA metabolizers showed an overall lower PCC than those of the three monophasic 4-AA metabolizers. The biphasic reduction of 4-AA in five calves suggests rapid distribution and subsequent redistribution from well-perfused organs based on the hypothesis that 4-AA has the ability to cross the blood-brain barrier and that the initial high amount of 4-AA in these calves is distributed to the central nervous system (CNS) [44]. From the CNS, it is subsequently redistributed into the blood, forming a second increase in plasma concentration. In contrast, small amounts of 4-AA generated in the other three calves seem to slowly accumulate in the intravascular compartment until being eliminated by renal routes, forming a monophasic elimination curve. The plasma 4-AA concentration was lower than that of 4-MAA in anesthetized calves. This implies that 4-MAA might not be completely converted into 4-AA, or peak 4-MAA concentration following administration of metamizole IV might have saturated the metabolic pathway of 4-MAA to 4-AA, metabolizing (oxidizing) a proportion of 4-MAA to 4-FAA [23]. According to the studies of Giorgi et al. [23] in horses and Aupanun et al. [21] in donkeys we do not know which metabolites exactly generate analgesic action in the calf. In humans, the analgesic effect of metamizole correlates with the concentrations of 4-MAA and 4-AA, which differ with regard to their time of onset (4-MAA > 4-AA) and terminal half-life (4-MAA: 4–5 hours, 4-AA: 5–8 hours) [11]. 4-MAA is approximately 50 times more active than metamizole as an inhibitor of the COX-3 enzyme [11], while 4-AA is less active. Therefore, both metabolites may contribute to the clinically relevant features of rapid onset and duration of the effect. The half-life of 4-MAA, however, is dose-dependent [45]. The other two metabolites 4-FAA and 4-AAA are inactive.

In humans, both active metamizole metabolites are known as direct and reversible COX inhibitors. Of course, compared to the described COX-1/2-relevant IC50 values (concentration of an inhibitor required to block a target *in vitro* to 50%) [46], in calves, only the plasma concentration of 4-MAA, but not of 4-AA, exceeded the COX-1/2-relevant IC50 values over a period of 24 hours. On the other hand, considering the reduced PCC values of the MGs compared to those of the CGs, one must assume a certain analgesic effect in our 5 biphasic 4-AA metabolizers. One could speculate here about an alternative analgesic mechanism to that of COX inhibition. After distribution into the brain and spinal cord, 4-AA may be transformed into CNS-restricted metabolites, which are potent regulators of the endocannabinoid and vanilloid systems [47, 48]. As both systems modulate inflammatory nociception [49, 50], CNS-distributed 4-AA might be relevant for enhanced suppression of perioperative pain in those animals.

While an analgesic effect of 4-MAA by inhibition of prostaglandin synthesis is likely in calves, interference with the preapplied COX inhibitor meloxicam is negligible, as meloxicam inhibits COX activity by competing with the arachidonic acid binding side, whereas 4-MAA by sequestering COX stimulating radicals [13]. As expected, preapplication of meloxicam did not prevent 4-MAA synthesis in calves in principle as 4-MAA generation occured within 15

minutes after application. The NSAID meloxicam was primarily given to produce postoperative analgesia as postoperative pain results from prostaglandin-induced inflammatory processes in response to surgical trauma [51]. Although meloxicam was already applied preemtively, a sufficient release of surgery-related inflammatory pain by NSAIDs is unclear [2].

Coetzee et al. [52] investigated the pharmacokinetics of intravenous and oral meloxicam application in ruminant Holstein calves without additional anesthesia. After intravenous administration (0.5 mg/kg, the same dose as our calves were administered), meloxicam demonstrated a relatively small mean apparent volume of distribution at steady state (Vss) of 0.171 L/kg (0.15–0.19 L/kg) and a slow clearance (Cl) from the central compartment of 0.1 ml/min/kg (0.08–0.12 ml/kg/min). This resulted in a relatively long mean plasma terminal half-life of 20.35 hours. However, anesthesia and especially changes in cardiac output may affect not only the distribution of a drug but also its elimination clearance if it has a high hepatic extraction ratio. Changes in the pharmacokinetics of a drug resulting from changes in cardiac output may affect both early and steady-state arterial drug concentrations as well as its context-sensitive half-times [53]. For example, Waterman [5] could already prove in calves that after premedication with xylazine, while not affecting the half-lives significantly, reduced volumes of distribution and the clearance rate of ketamine and norketamine plasma concentrations by up to 50% (in females) compared to unpremedicated calves. As we have no control group of unanesthetized metamizole-treated calves included in our study, our results are probably not generally valid for nonanesthetized animals. Additionally, there was also no detailed pharmacokinetic investigation of drug-drug interference included in the current study. Theoretically, drug interference with ketamine and xylazine may occur not only by substrate competition due to the previously mentioned CYP3A liver enzyme metabolism but also by influencing the cardiovascular state of the animal [54]. Our already published clinical monitoring data [27] showed that the MAP significantly increased during anesthesia, independent of metamizole administration. This may have resulted from a waning in the hypotensive action of xylazine [55]. The HR initially decreased in almost parallel courses with and without metamizole until the beginning of the 'suturing of the peritoneum and fascia' period (29 minutes after skin incision), probably related to the depressive casrdiovascular effect of ketamine-xylazine, before it remained relatively constant until the end of surgical intervention with metamizole as an add-on. The CGs, in contrast, showed an increase in HR after beginning of the 'suturing of the peritoneum and fascia' period.

## Conclusion

Preliminary metabolic behavior data showed that presurgical applied metamizole is rapidly transformed into the analgesic metabolites 4-MAA and 4-AA in calves subjected to umbilical surgery under general anesthesia with ketamine, xylazine and isoflurane. Preapplied meloxicam did not obviously interfere with metamizole conversion. In conjunction with previously published data on HR, MAP and PCCs in the same animals, metamizole impairs nociception for at least 8 hours after application. Thus, metamizole is a promising candidate for preemptive analgesia in calves and might be recommended for improving perioperative pain management. It remains to be evaluated whether different kinetics of 4-AA synthesis are de facto of pharmacological relevance.

## Acknowledgments

The authors would like to thank the Bremen Medical Laboratory (MLHB) for performing the 4-MAA/4-AA analysis, especially Prof. W.N. Kühn-Velten and Dr. G. Zurek.

## Author Contributions

**Conceptualization:** Moritz Metzner, Christine Baumgartner.

**Data curation:** Daniela Fux, Magdalena Behrendt-Wippermann.

**Formal analysis:** Daniela Fux, Johanna Brandl.

**Funding acquisition:** Magdalena Behrendt-Wippermann.

**Investigation:** Moritz Metzner, Melanie Feist, Magdalena Behrendt-Wippermann.

**Methodology:** Moritz Metzner, Melanie Feist, Anne von Thaden, Christine Baumgartner.

**Project administration:** Moritz Metzner, Magdalena Behrendt-Wippermann, Christine Baumgartner.

**Supervision:** Moritz Metzner, Christine Baumgartner.

**Validation:** Daniela Fux, Christine Baumgartner.

**Writing – original draft:** Daniela Fux, Moritz Metzner, Johanna Brandl.

**Writing – review & editing:** Daniela Fux, Moritz Metzner, Johanna Brandl, Melanie Feist, Magdalena Behrendt-Wippermann, Anne von Thaden, Christine Baumgartner.

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
