## [Decision Letter · Decision Letter 0]

30 Dec 2020

PONE-D-20-36269

Pharmacokinetics of metamizole (dipyrone) used as an add on analgesic in calves undergoing umbilical surgery

PLOS ONE

Sehr geehrte Frau Dr. Baumgartner,

Thank you for submitting your manuscript to PLOS ONE. After careful consideration, we feel that it has merit but does not fully meet PLOS ONE’s publication criteria as it currently stands. Therefore, we invite you to submit a revised version of the manuscript that addresses the points raised during the review process.

Both expert reviewers are supportive of this study. However, they have identified several aspects that need to be adequately addressed for this manuscript to be considered further. I concur with their view and believe that you will find their comments very helpful for improving the manuscript. I look forward to receiving your revised version.

We look forward to receiving your revised manuscript.

Kind regards,

Angel Abuelo, DVM, MRes, MSc, PhD, DABVP (Dairy), DECBHM

Academic Editor

PLOS ONE

P.S.  Bitte begrüßen Sie Dr. Metzner in meinem Namen!

Journal Requirements:

2. In your Methods section, please provide additional details regarding the animals used in your study and ensure you have described the source. For more information regarding PLOS' policy on materials sharing and reporting, see https://journals.plos.org/plosone/s/materials-and-software-sharing#loc-sharing-materials.

3. In your Methods section, please state the volume of the blood samples collected for use in your study.

4. We noted you reported at line 105 "the animals were sold afterwards to a local cattle dealer". As we understand that the animals were sold for food purposes at the end of the study, please confirm whether this was approved by the relevant national and/or local Food Safety Administration.

5.Thank you for stating the following in the **Financial Disclosure** section:

"Magdalena Behrendt-Wippermann got a grant  from the H. Wilhelm Schaumann Stiftung Germany (https://www.schaumann-stiftung.de/de/forderung-1764.htm).  The study was additionally supported by Richter Pharma AG, Austria (https://www.richter-pharma.at/).

We note that you received funding from a commercial source: Richter Pharma AG

Reviewers' comments:

Reviewer's Responses to Questions

**Comments to the Author**

1. Is the manuscript technically sound, and do the data support the conclusions?

Reviewer #1: Partly

Reviewer #2: Partly

2. Has the statistical analysis been performed appropriately and rigorously? 

Reviewer #1: No

Reviewer #2: I Don't Know

3. Have the authors made all data underlying the findings in their manuscript fully available?

Reviewer #1: No

Reviewer #2: Yes

4. Is the manuscript presented in an intelligible fashion and written in standard English?

Reviewer #1: No

Reviewer #2: Yes

5. Review Comments to the Author

Reviewer #1: Pharmacokinetics of metamizole (dipyrone) used as an add on analgesic in calves undergoing umbilical surgery. Fux D, et al., 2020

Interesting study with a combination of analgesic drugs, other drugs. There is a disconnect between the actual study and the description of similarities and differences between models. Much to consider. My suggestion is to revise (shorten) to make flow more appropriate. If you meant to conduct PK-PD study, I think it missed the mark. If you want to conduct PK study in calves, describe this study. Some attention to possible implications may be helpful, but limit to ruminant or monogastric etc.

Abstract –

The first line of the abstract states: The pharmacokinetic behaviour of metamizole (dipyrone) and its metabolites 4-MAA, 4-AA was evaluated in this clinical, veterinary research study in calves…

While the statements below indicate only 4-MAA and 4-AA were followed (based upon rapid conversion of metamizole to metabolites). However, it does not appear plasma concentration of parent drug were followed. Please revise to make this correct.

Line 34. “time point 15 minutes after application.” If the drug was topically applied, application is an appropriate term. If given parenterally, administration/injection/IV catheter etc is likely more appropriate.

Line 36-39. Intriguing finding of different disposition for 4-AA. Please expand upon this information?

Line 42-48. You indicate that metamizole is converted to 4-MAA and 4-AA in anesthetized calves. You then state 4-MAA follows 2-compartment model with rapid distribution and slower elimination phases. They you state that kinetic data support the clinical analgesic effect of observed there. You do not provide any data to support the analgesic effects of metamizole? Please clarify how Tmax, Cmax, for metabolites demonstrates analgesic effects?

Line 62-63. Please clarify how one can tease the 4-MAA, 4-AA effects of metamizole from the systemic effects of meloxicam (co-administered to calves)? Did the meloxicam produce changes in disposition of 4-MAA, 4-AA or is there a pharmacogenetic component in these animals?

Line 70-71. Labelled for humans, cats, dogs, horses, cattle, swine. From the US FDA (https://www.fda.gov/animal-veterinary/product-safety-information/zimeta-dipyrone-injection-veterinarians), the following is stated:

Zimeta belongs to the pyrazolone class of non-steroidal anti-inflammatory (NSAID) drugs and is the first injectable dipyrone product to receive FDA approval for use in horses. Zimeta is for use in horses only. Zimeta has not been evaluated for use in horses intended for human consumption or food-producing animals, including lactating dairy animals.

Zimeta is not for use in humans. Direct contact with the skin should be avoided. Precautions should be employed by practitioners when handling and using loaded syringes to prevent accidental self-injection, as epidemiological studies have indicated that dipyrone can cause agranulocytosis in humans.

Perhaps this should be clarified (the FDA product safety information) is specific to the USA. They also use language such as: “Dipyrone is now prohibited for use in humans in several countries.” Similar language may clarify this difference in countries where the use is not controlled as much?

Line 132-134. Was the surgery time 52 minutes in each animal? If not, some measure of variability in surgery time should be provided (± SD).

Line 164-166. Mean is generally paired with std deviation. Median is generally paired with range?

Line 136 – 153. Was there any interference between 4-MAA, 4-AA and meloxicam, xylazine, ketamine (chromatography, MS)? Why not use deuterated compound for IS?

Line 222-223 – AUC is correctly represented as ug*mL/hour?

I suggest that it would have been prudent to use half of the calves to receive metamizole and the other half do not get the same meloxicam. I would also suggest the authors attempt to shorten the discussion to make the paper clearer (emphasizing the important points?

For example:

Line 205-208 – How do the authors know the Cmax occurred at different time point in calves vs. dairy cows? Were the routes of administration the same? Was the dose of drug the same? It is clear that the drug used in each study was not the same, and interactions may occur?

Line 233-242. Please clarify: 303 ug/mL 4-MAA in calves vs. 101.62 ug/mL in adult cattle was a third (in calves) less than adult? Please clarify. If 4-MAA is only 1/3 of cows, then next sentence says 22.6 mg/kg vs. 40 mg/kg. Is this higher in calves and lower in cows or vice versa?

Line 239-241. What is the importance of higher drug level in calf than in adult cow? Neonates tend to be underdosed compared with adults. Depends upon species. Could also depend upon breed?

There is much more information in the discussion that discredits this manuscript. Please shorten, provide concise information about calves. Spend small amount of time discussing importance of breed, age, sex etc. on PK of drug. The kinetics of drugs and their metabolites tells little about drug effect (without additional data). Consider this when describing the results etc.

I think one of the strengths of this paper is the clinical approach. However, judicious use of drugs is helpful. Further, having a control and test group would help?

Reviewer #2: As there is not a lot of information in the literature on the pharmacokinetics of dipyrone/metamizole in cattle, this study has the potential to better inform clinical veterinary practice and NSAID decision-making. I have a few general and specific comments that may help improve the manuscript.

Title:

Since analgesia was not specifically evaluated in the study, it would be best to remove the word from the title. In the discussion, however, it would still be appropriate to speculate on the analgesic potential based on achievable plasma concentrations.

Abstract:

Line 42: It might be more appropriate to say that those metabolites were present, but there may be others that were not measured. The way the sentence currently reads suggests that those two metabolites are the only ones. Also, a better way to describe the model fit would be "observed data appeared to fit a two-compartment model best", rather than "corresponded to", since the method of analysis should be to attempt various models.

Introduction:

It seems important to mention here and in the discussion section that this drug has been associated with agranulocytosis in humans and has been banned for use in food animals at various times in recent years in some countries. The food safety aspect of using it ought to be at least mentioned (e.g., is there an MRL/tolerance established, have withdrawal times been reported?). This is briefly mentioned in the introduction, but a bit more discussion might be appropriate, and naming the countries in which it is approved in food animals might be helpful to the reader. In addition, if approved in cattle, what indications were approved, and what dose, since this might help the reader put this study in context.

Line 87: what dose was used in the previously published study?

Methods:

Given the reports of agranulocytosis in people, it would be helpful if the investigators reported the reasons why CBCs were not performed at any time during the study, especially after drug administration.

Line 147: Thank you for including LOD and LOQ. It would be helpful to note if the samples were all analyzed on the same day, and if not, was interday variability evaluated? This is not essential, but would be helpful for the reader.

Line 155: It would be appropriate for PK analysis to include attempts at fitting at least one- and two-compartment models as well as the method used to select which model. Please also clarify whether the parameters mentioned in the first sentence are from observed or modeled data. Were models attempted for the 4-AA data and they would not converge? It would be helpful to include that information, as an addition to explaining why only noncompartmental analysis was performed for this metabolite.

Results:

The pharmacokinetic parameter data would be easier to review in tabular form. In addition, I am curious as to why the two-compartment model was described but the estimated parameters were not reported.

Additional consideration should be given as to the precision of the parameter estimates and significant figures. Reporting 100ths of hours, for example, is likely not meaningful and is an artifact of the modeling software and not the precision of the estimate.

Discussion:

The discussion could do with a bit of reorganization to group like items closer together, for example, discussion of observed plasma concentrations and the observed PK parameters , followed by discussion of metabolites, and then the potential for drug interactions with the anesthetics. Also, I’m not sure comparing with non-ruminants is helpful if there are some data in ruminant species.

It is important to speculate on the effects on clearance and volume of distribution during anesthesia, particularly comparing with other drugs in cattle with and without the additional drugs.

Line 206: The difference between 15 and 30 minutes is not likely to be meaningful, since samples were collected in either study only at those time points and not in between.

Line 212: It might be more appropriate to state that "it is assumed that the animals were not anesthetized and no other drugs were given at the same time".

Line 213: This sentence needs some clarification: are you suggesting there are drug-drug interactions? If so, perhaps be more explicit and provide more discussion than just the reference.

Line 223: Has the esterase been demonstrated to be the metabolic pathway in cattle? Perhaps this discussion should be more speculative given the lack of direct information, or additional evidence to support this based on the type of metabolite.

Line 310: As previously stated, care should be taken to make anything significant out of the differences in Cmax. In addition, the clinical relevance of any differences seems low, since the important parameter for analgesia is more likely to be time above an inhibitory concentration rather than a peak concentration.

Line 318: CYP enzymes are species-specific, so caution should be taken to make that clear in this part of the discussion. What is known about the isoforms in cattle and what is speculation vs conclusive?

Line 337: Description of the methods of analgesia determination in the previous study, with perhaps more details about the conclusions, should be at least briefly included here. In addition, there are not sufficient data or analysis in this study to define a clear PK-PD relationship, so this should be quite speculative, and the impact of the other drugs used on analgesia has not been clearly delineated.

Figures:

Drug concentration data and the visual assessment of linearity is more easily done when the x-axis is complete rather than broken. Consideration might also be given to a log10 scale for the y-axis, again for assessment of linearity.

6. PLOS authors have the option to publish the peer review history of their article (what does this mean?). If published, this will include your full peer review and any attached files.

Reviewer #1: No

Reviewer #2: No

---

## [Author Response · Author response to Decision Letter 0]

28 Jun 2021

Thank you very much for your supportive comments and the opportunity to resubmit our manuscript. Please find our answer to you comments in our attached file labeled 'Response to Reviewers'. We very much hope to have improved the aspects raised by you to your satisfaction.

---

## [Decision Letter · Decision Letter 1]

28 Jul 2021

PONE-D-20-36269R1

Pharmacokinetics of metamizole (dipyrone) used as an add-on in calves undergoing umbilical surgery

PLOS ONE

Dear Dr. Baumgartner,

Thank you for submitting your manuscript to PLOS ONE. After careful consideration, we feel that it has merit but does not fully meet PLOS ONE’s publication criteria as it currently stands. Therefore, we invite you to submit a revised version of the manuscript that addresses the points raised during the review process.

Thank you for your efforts in revising the manuscript. Both reviewers and I acknowledge that the manuscript has been much improved. Nevertheless, there are still a few important points raised by reviewer #2 that would need to be addressed before a recommendation for acceptance can be made. Please, as suggested by the reviewer, also enlist the help of a colleague with experience in academic English language. While the grammar and syntax of the manuscript is overall correct, there are instances were language results awkward (probably as a result of direct translation from German to English). PLoS One does not provide language editing during typesetting and, therefore, I strongly suggest to have this taken care of now.

We look forward to receiving your revised manuscript.

Kind regards,

Angel Abuelo, DVM, MRes, MSc, PhD, DABVP (Dairy), DECBHM

Academic Editor

PLOS ONE

Reviewers' comments:

Reviewer's Responses to Questions

**Comments to the Author**

1. If the authors have adequately addressed your comments raised in a previous round of review and you feel that this manuscript is now acceptable for publication, you may indicate that here to bypass the “Comments to the Author” section, enter your conflict of interest statement in the “Confidential to Editor” section, and submit your "Accept" recommendation.

Reviewer #1: All comments have been addressed

Reviewer #2: (No Response)

2. Is the manuscript technically sound, and do the data support the conclusions?

Reviewer #1: Yes

Reviewer #2: Partly

3. Has the statistical analysis been performed appropriately and rigorously? 

Reviewer #1: Yes

Reviewer #2: Yes

4. Have the authors made all data underlying the findings in their manuscript fully available?

Reviewer #1: Yes

Reviewer #2: Yes

5. Is the manuscript presented in an intelligible fashion and written in standard English?

Reviewer #1: Yes

Reviewer #2: No

6. Review Comments to the Author

Reviewer #1: Thank you for addressing my concerns. A very exhaustive revision that should greatly improve the overall study.

Reviewer #2: Because of the extensive revisions to the manuscript, it was difficult to evaluate only the revised portions, so I read it as if it were a new submission. It's possible, therefore, that there may be some repeated comments or suggestions. I make the following suggestions and comments to hopefully strengthen this submission.

LANGUAGE: I recommend that the authors have the manuscript reviewed and edited by a native English speaker, as there are instances of awkward phrasing or inappropriate word use (for example, pain is not "released" by an analgesic, it is eliminated or reduced by one).

GENERAL COMMENTS: As previously recommended by a reviewer, this manuscript should tighten up the focus to a straight forward description of the pharmacokinetics of two of the known metabolites of metamizole. It should also reduce the amount of speculation in the discussion about analgesia, pharmacodynamics, and metabolic differences among populations of cattle, given that this investigation did not evaluate any of those. A small amount of speculation is certainly warranted, e.g., plasma concentrations reported in other species to be analgesic, or CYP450 isoforms associated with differences in metabolism, but they should be short and succinct. In addition, the impact of the sampling times on the parameter estimates is not discussed at all, and there is a possibility that they might have impacted the results and their interpretation.

OTHER SPECIFIC SUGGESTIONS:

ABSTRACT: One cannot describe two Cmax's - there is only one "highest" which is what Cmax represent. In addition, the time of Cmax is properly called Tmax, rather than conflating the two parameters into a single report with time and concentration. Describing two peaks is reasonable, and speculating about the reason is as well, but they are not both Cmax.

Line 91 and others: Please use the phrase withdrawal time, as it is understood among cattle health professionals what that means. (Avoid "latency" or "waiting period".)

Line 195: One cannot collect blood with EDTA and extract serum - please clarify whether plasma or serum was used to analyze drug concentrations.

Line 212/PK analysis: More details about the two-compartment model selected are needed (even if it's just to describe the model number within the software), as well as why it was selected - it's hard to believe one-compartment didn't converge at all, but perhaps the AIC and visual examination of the residuals suggested two-compartment was a better fit? In addition, noncompartmental analysis only describes what's there, so there is no correlation needed to determine if it's appropriate.

Line 223/Results - PK parameters are best provided in a table, and there are some parameters missing from the results, such as the micro-constants. One cannot evaluate PK just from figures representing serum concentrations, so tabular data is expected. In addition, given the length of the estimated half-lives, and that the duration of sample collection was only 24 hours, the authors will want to speculate in the discussion about whether samples were collected for long enough to actually effectively estimate half-life and other parameters (was extrapolation to infinity possible for example).

7. PLOS authors have the option to publish the peer review history of their article (what does this mean?). If published, this will include your full peer review and any attached files.

Reviewer #1: **Yes: **Jeffrey Lakritz

Reviewer #2: No

---

## [Author Response · Author response to Decision Letter 1]

19 Oct 2021

Thank you very much for the opportunity to resubmit again our manuscript to PLOS ONE after careful revision. We have had the manuscript professionally edited by the American Journal Experts (AJE) (see certificate attached). We hope that the linguistic improvement now meets your expectations. Please find our detailed answers to your questions in the attached file labeled 'Response to Reviewers'.

---

## [Decision Letter · Decision Letter 2]

28 Oct 2021

PONE-D-20-36269R2Pharmacokinetics of metamizole (dipyrone) as an add-on in calves undergoing umbilical surgeryPLOS ONE

Dear Dr. Baumgartner,

Thank you for submitting your manuscript to PLOS ONE. After careful consideration, we feel that it has merit but does not fully meet PLOS ONE’s publication criteria as it currently stands. Therefore, we invite you to submit a revised version of the manuscript that addresses the points raised during the review process. Please address the comments of reviewer #1 through another round of revisions.

We look forward to receiving your revised manuscript.

Kind regards,

Angel Abuelo, DVM, MRes, MSc, PhD, DABVP (Dairy), DECBHM

Academic Editor

PLOS ONE

Journal Requirements:

Reviewers' comments:

Reviewer's Responses to Questions

**Comments to the Author**

1. If the authors have adequately addressed your comments raised in a previous round of review and you feel that this manuscript is now acceptable for publication, you may indicate that here to bypass the “Comments to the Author” section, enter your conflict of interest statement in the “Confidential to Editor” section, and submit your "Accept" recommendation.

Reviewer #1: (No Response)

Reviewer #2: All comments have been addressed

2. Is the manuscript technically sound, and do the data support the conclusions?

Reviewer #1: Partly

Reviewer #2: Yes

3. Has the statistical analysis been performed appropriately and rigorously? 

Reviewer #1: No

Reviewer #2: Yes

4. Have the authors made all data underlying the findings in their manuscript fully available?

Reviewer #1: Yes

Reviewer #2: Yes

5. Is the manuscript presented in an intelligible fashion and written in standard English?

Reviewer #1: No

Reviewer #2: Yes

6. Review Comments to the Author

Reviewer #1: PONE-D-20-36269R2 - Pharmacokinetics of metamizole (dipyrone) as an add-on in calves undergoing umbilical surgery

Line 70-72. Please cite references for this comment:

“However, the rare risk of reversible but potentially fatal 71 agranulocytosis led to the introduction of compulsory prescriptions for metamizole in Germany.”

I would want to know if residues in animals may be harmful to consumers

Line 132-133. Please revise this sentence. As written, the meaning is nonsensical

“The clinical, veterinary research study was conducted between August 2013 and July 2014, except for wintertime, during which the temperature was too low.”

Indicate which months were excluded and also please define why temperature is too low (cold enough to result in damage to the active ingredient?)

Line 28-29. Please explain why you could not evaluate drug in young animals that are anesthetized. Since these are milk fed animals with hay, TMR, with mineral lick availability (TMR does not include minerals?). Pre-ruminants may have markedly different disposition of this drug when compared to ruminants? Including a statement regarding quantity of hay/TMR consumed daily may help.

Line 146-150. I think I would indicate all calves were purchased to repair hernia? Then state all were healthy during the study?

Line 162-166. It would be prudent to indicate whether drug was administered by the same catheter as blood sampling for analysis of metamizole metabolites. If 2 catheters placed, then indicate so in methods section.

Line 190-210. I think you should consider providing accuracy, precision for both individual runs as well as between day runs. Linearity is another consideration.

While we are on subject of quantification, how does one determine T1/2 elimination, clearance, Vd etc of metabolite(s) after injection of the parent drug? How does one get Vss of 17.76 from the 5 rapid and 3 slow metabolizers (249.19 + 283.36/2)? If the Volumes for metabolites is so large, why does clearance increase for slow metabolizers. Why is Vss units of mg/(ug/mL)

Line 285-289. I am confused about the stated goals of this study. Apparently you have data on plasma cortisol concentrations after metamizole and surgery/anesthesia. I do not see how you can comment on this without having controls. It is very confusing you did not specifically address the changes in PCC with metamizole in calve undergoing anesthesia and surgery. Without your stated controls, this study seems woefully incomplete.

Reviewer #2: (No Response)

7. PLOS authors have the option to publish the peer review history of their article (what does this mean?). If published, this will include your full peer review and any attached files.

Reviewer #1: No

Reviewer #2: No

---

## [Author Response · Author response to Decision Letter 2]

7 Dec 2021

Thank you very much for the opportunity to resubmit our manuscript to PLOS ONE after minor revision. Please find our detailed answers to your comments in the attached file labeled 'Response to Reviewers'.

---

## [Decision Letter · Decision Letter 3]

21 Dec 2021

PONE-D-20-36269R3Pharmacokinetics of metamizole (dipyrone) as an add-on in calves undergoing umbilical surgeryPLOS ONE

Dear Dr. Baumgartner,

Thank you for submitting your manuscript to PLOS ONE. After careful consideration, we feel that it has merit but does not fully meet PLOS ONE’s publication criteria as it currently stands. Therefore, we invite you to submit a revised version of the manuscript that addresses the points raised during the review process. In line with the reviewer's comment, you must provide the details/results of the method validation in the manuscript, even if the analyses were conducted at an external laboratory.

We look forward to receiving your revised manuscript.

Kind regards,

Angel Abuelo, DVM, MRes, MSc, PhD, DABVP (Dairy), DECBHM

Academic Editor

PLOS ONE

Journal Requirements:

Reviewers' comments:

Reviewer's Responses to Questions

**Comments to the Author**

1. If the authors have adequately addressed your comments raised in a previous round of review and you feel that this manuscript is now acceptable for publication, you may indicate that here to bypass the “Comments to the Author” section, enter your conflict of interest statement in the “Confidential to Editor” section, and submit your "Accept" recommendation.

Reviewer #1: All comments have been addressed

2. Is the manuscript technically sound, and do the data support the conclusions?

Reviewer #1: Partly

3. Has the statistical analysis been performed appropriately and rigorously? 

Reviewer #1: I Don't Know

4. Have the authors made all data underlying the findings in their manuscript fully available?

Reviewer #1: Yes

5. Is the manuscript presented in an intelligible fashion and written in standard English?

Reviewer #1: Yes

6. Review Comments to the Author

Reviewer #1: It would be appropriate to provide method validation details for this type of study (linearity, within and between day variability, stability, etc.)

7. PLOS authors have the option to publish the peer review history of their article (what does this mean?). If published, this will include your full peer review and any attached files.

Reviewer #1: No

---

## [Author Response · Author response to Decision Letter 3]

24 Feb 2022

We very much hope that we have now sufficiently answered your last request. Please find our detailed answers in the attached file labeled 'Response to Reviewers'.

---

## [Editor Report · Decision Letter 4]

1 Mar 2022

Pharmacokinetics of metamizole (dipyrone) as an add-on in calves undergoing umbilical surgery

PONE-D-20-36269R4

Dear Dr. Baumgartner,

We’re pleased to inform you that your manuscript has been judged scientifically suitable for publication and will be formally accepted for publication once it meets all outstanding technical requirements.

Kind regards,

Angel Abuelo, DVM, MRes, MSc, PhD, DABVP (Dairy), DECBHM

Academic Editor

PLOS ONE
---

## [Editor Report · Acceptance letter]

4 Mar 2022

PONE-D-20-36269R4 

Pharmacokinetics of metamizole (dipyrone) as an add-on in calves undergoing umbilical surgery 

Dear Dr. Baumgartner:

I'm pleased to inform you that your manuscript has been deemed suitable for publication in PLOS ONE. Congratulations! Your manuscript is now with our production department. 

Kind regards, 

on behalf of

Dr. Angel Abuelo 

Academic Editor

PLOS ONE